# Sleep Deprivation and Central Appetite Regulation

**DOI:** 10.3390/nu14245196

**Published:** 2022-12-07

**Authors:** Shuailing Liu, Xiya Wang, Qian Zheng, Lanyue Gao, Qi Sun

**Affiliations:** 1Department of Child and Adolescent Health, School of Public Health, China Medical University, Shenyang 110122, China; 2Experimental Center for School of Public Health, China Medical University, Shenyang 110122, China

**Keywords:** sleep deprivation, central appetite, animal models, circadian, childhood obesity

## Abstract

Research shows that reduced sleep duration is related to an increased risk of obesity. The relationship between sleep deprivation and obesity, type 2 diabetes, and other chronic diseases may be related to the imbalance of appetite regulation. To comprehensively illustrate the specific relationship between sleep deprivation and appetite regulation, this review introduces the pathophysiology of sleep deprivation, the research cutting edge of animal models, and the central regulatory mechanism of appetite under sleep deprivation. This paper summarizes the changes in appetite-related hormones orexin, ghrelin, leptin, and insulin secretion caused by long-term sleep deprivation based on the epidemiology data and animal studies that have established sleep deprivation models. Moreover, this review analyzes the potential mechanism of associations between appetite regulation and sleep deprivation, providing more clues on further studies and new strategies to access obesity and metabolic disease.

## 1. Introduction

As the use of artificial light sources has significantly increased, the severity of insufficient sleep has increased in parallel [1]. Sleep duration curtailment is most common in adolescents facing high pressure in their school performances, and young people who frequently work overtime or conduct shift work [2,3]. In recent years, sleep duration curtailment has become an increasingly serious problem for children and adolescents globally. In 2015, a report from the Centers for Disease Control and Prevention in the United States on adolescent risk behavior surveillance showed that 57.8% of middle school students and 72.7% of high school students did not have enough sleep [4,5]. Insufficient sleep can lead to diminished abstract thinking and memory, low energy, and seriously affect the growth and development of children [6]. Even more noteworthy, children who have less sleep have a higher chance of being overweight or obese in the future. According to a prospective cohort study, children aged 2.5 to 6 years who slept less than 10 h had a 2.9 times higher risk of obesity than those who slept more than 11 h [7]. Another birth cohort study analyzed the relationship between the sleep duration of children and body mass index (BMI) in the adulthood stage, showing that children who slept less had a greater BMI in adulthood [8].

According to World Health Organization survey data, the number of obese people worldwide has nearly tripled since 1975. By 2016, more than 340 million children and adolescents aged 5–19 years were overweight or obese, and their prevalence had increased dramatically from only 4% in 1975 to more than 18% in 2016 [9]. If the current trend continues, by 2030, nearly 60% of the world’s population will be overweight or obese [10]. Therefore, identifying causative factors is important for an effective response to the obesity epidemic.

In addition to genetic factors, changes in individual lifestyles and the environment have significantly increased the obesity rate in the last 20 years. The causality of sleep curtailment and obesity can be attributed to neuroendocrine changes, which are related to appetite disorders [11,12]. The central mechanisms underlying the effects of sleep deprivation on appetitive food desires, which might result in weight gain, remain unexplained, as population-level and peripheral body data have pointed out. The hypothalamus is the senior center under the cerebral cortex that regulates visceral activity, linking it to other physiological activities and regulating important physiological functions, such as biorhythms, feeding, water balance, body temperature, and endocrine gland activity [13]. The hypothalamus is a key part of the central nervous system that integrates various appetite regulation signals and maintains body weight [14]. An imbalance in central appetite regulation is a key event in the development of obesity [15]. In this review, we synthesize epidemiological studies and animal studies of sleep deprivation to explore the links and mechanisms between sleep deprivation and appetite regulation. Firstly, we present the pathophysiology of sleep deprivation and its role in appetite regulation functions and focus on the relationship between sleep deprivation and a number of appetite-regulating hormones, including the melanocortin system, ghrelin, orexin, leptin, and insulin. The mechanisms that may influence changes in the levels of these hormones will be presented. Secondly, we compared various sleep deprivation models based on the type of experimental animal, age, experimental design, length of sleep deprivation, and indications of the outcomes. We highlight the need for sleep deprivation to be studied in animal experiments and explore the potential implications of sleep treatments for obese populations.

## 2. Sleep Deprivation

### 2.1. Pathophysiology of Sleep Deprivation

The term “sleep deprivation” refers to “abnormal sleep conditions that exhibit deficient sleep quantity, structure, and/or quality” [16]. Chronic sleep deprivation has significant adverse effects on health and overall quality of life, and individuals with chronic sleep deprivation have significantly lower quality of life scores [17]. Chronic sleep deprivation is associated with elevated cortisol and decreased testosterone levels [18]. Elevated cortisol levels are known to be associated with depression, anxiety, hypertension, obesity, and type 2 diabetes [19]. In addition, chronic sleep deprivation is associated with increased inflammatory markers, which are associated with psychiatric disorders [20]. Chronic sleep deprivation could potentially lead to the development of other diseases since numerous physiological changes occur in the body during chronic sleep deprivation. It is critical for patients with physiological or psychological disorders to obtain enough high-quality sleep.

### 2.2. Animal Model of Sleep Deprivation

To investigate the relationship between sleep deprivation and food intake, as well as body weight and metabolism, researchers have performed studies by establishing animal models of sleep deprivation (Table 1). Different sleep deprivation methods that were summarized in this table were adopted depending on the animal models. There are four broad categories: platform technique, forced locomotion technique, gentle handling, and biological intervention method. Among them, the platform method was proposed by Cohen et al. [21] in 1966 to induce rapid eye movement (REM) sleep deprivation by using the water-fearing characteristics of rats. The rat was placed onto a platform surrounded by water 1 cm below the platform; when the experimental rat entered the REM sleep period, nodding occurred with muscle relaxation, then the body fell into the water; hence, the rat woke up and remained awake. The forced movement method is usually performed in a space containing a rotating axis or running wheel. The experimental animal needed to keep walking to avoid touching or falling down. One common disadvantage of both two methods is that the effects of environmental disturbances can cause stress to the experimental animal. The gentle handling method requires the experimenter to gently touch the whiskers or fur, or tap the cage wall to keep the experimental animal awake when the experimenter observes the animal entering a sleep state to achieve the effect of sleep deprivation. The disadvantage of this method is that the artificial recognition of sleep state is prone to error. Finally, the biological intervention method usually involves the destruction of specific areas of the brain or special genetic programming to achieve sleep deprivation [22]. On the other hand, there are different deprivation programs to be studied regarding the sleep-deprived duration. It is interesting that the researchers obtained different body weight outcomes in these experimental animals, even though the food intakes were increased consistently across all sleep deprivation models. They are broadly divided into acute continuous sleep deprivation and chronic intermittent sleep deprivation. The construction of the above models has given researchers the opportunity to explore in depth the mechanisms by which sleep affects biological behavior, thus providing theoretical support for clinical interventions.

## 3. Sleep Curtailment and Appetite Regulation in Human

There is growing evidence that reduced sleep duration in children is associated with an increased risk of being overweight and obese later in life. A meta-analysis confirmed a significant association between short sleep duration and adverse changes in body mass index (BMI) in infants, children, and adolescents [35,36,37], and quantified the associated risk of a greater risk of overweight/obesity in children with short sleep durations. The findings further support the existence of a positive association between reduced nighttime sleep duration and childhood obesity. Additionally, the results of interventions targeting sleep suggest that improved sleep duration or quality may be beneficial in reducing weight gain in children [38]. Notably, the results of the experiments in healthy adult volunteers found that food intake increased during sleep deprivation, providing the energy needed for additional wakefulness [39,40]. Conversely, switching from sleep deprivation to adequate/restorative sleep reduced energy intake, especially fat and carbohydrate components, and led to weight loss [41], indicating that sleep regulates body weight by influencing the balance between energy expenditure and intake and that exploring this balance would facilitate further interventions for childhood obesity. Interestingly, current research suggests that reduced sleep duration can affect a child’s ability to self-regulate his/her appetite and develop poor eating patterns, thereby increasing the risk of overeating [42,43]. Hjorth et al. [44] also demonstrated that a 1 h decrease in sleep duration increased the intake of added sugar and sugar-sweetened beverages. These indicate that investigating the effects of sleep deprivation on the appetite regulation of children would be useful for further elucidating the mechanisms underlying weight gain in children induced by sleep deprivation.

## 4. Mechanisms of Sleep Deprivation on Regulating Appetite

An increasing number of studies are investigating the relationship between sleep deprivation and appetite regulation, as well as their potential mechanisms, via sleep deprivation models. The hypothalamus is a key part of the central nervous system that integrates various appetite regulation signals and maintains body weight. An imbalance in central appetite regulation is a key event in the development of obesity [15]. Therefore, in the following section, we mainly discuss the central regulatory mechanisms of sleep deprivation on appetite.

### 4.1. Central Appetite Regulation

The hypothalamus has several neuronal centers, the lateral hypothalamic nucleus, which is considered the “hunger” center, and the ventromedial nucleus, which is the “satiety” center. In addition, the paraventricular nucleus and the hypothalamic arcuate nucleus (ARC) are sites where multiple hormones released from the gut and adipose tissue converge to regulate food intake and energy expenditure. Two different types of neurons in the ARC regulate appetite. The anorexigenic (appetite-suppressing) pro-opiomelanocortin (POMC) neurons, and the orexigenic (appetite-increasing) neuropeptide Y (NPY)/agouti-related peptide (AgRP) neurons [45]. The ARC integrates inputs from the vagus nerve and body fluids, including orexin, ghrelin, leptin, and insulin, and plays a physiological role in the regulation of appetite (Figure 1).

#### 4.1.1. The Role of the Melanocortin System in Central Appetite Regulation

In mammals, the melanocortin system is a key neuroendocrine network that plays a critical role in the regulation of appetite and energy homeostasis. The hypothalamic “melanocortin” neural loop is involved in the regulation of energy homeostasis. This loop is composed of appetite-suppressing POMC neurons in the hypothalamic arcuate nucleus, appetite-promoting AgRP neurons, melanocortin receptor-4 (MC4R)-positive neurons in the paraventricular nucleus, and the neural projections between them [46]. During satiety, POMC neurons project and release α-MSH, which activates MC4R on paraventricular nucleus neurons to reduce appetite, while during starvation, AgRP neurons send inhibitory GABAergic neural projections to suppress POMC neuronal excitability and can also secrete AgRP to antagonize the activation of MC4R by POMC, thus enhancing appetite. Thus, AgRP neurons, POMC neurons, and paraventricular nucleus neurons form an important neural loop to maintain the energy balance of the body [47].

#### 4.1.2. The Role of Orexin and Ghrelin in Central Appetite Regulation

Orexin was first discovered in 1998 in rat brain tissue extracts and is synthesized in the lateral hypothalamic region [48]. Orexin neurons are involved in the control of various homeostatic functions, including feeding and energy expenditure [49]. When injected intraperitoneally, orexin stimulates food intake [50]. Fasting then leads to an upregulation of orexin mRNA levels [51] and increases the number of excitatory synapses on orexin neurons [52]. Blocking orexin receptors reduces food intake [53] and binge-eating behavior [54].

Ghrelin release from neurons in the gastric oxyntic gland and the arcuate nucleus of the hypothalamus (ARC) stimulates growth hormone release and food intake [55] and can increase synaptic activity in NPY/AgRP neurons. This effect is achieved by activating an adenylate-activated protein kinase (AMP-activated protein kinase, AMPK)-dependent positive feedback loop, and the effects of ghrelin persist for several hours even after the removal of this kinase [56]. In addition, intravenous administration of ghrelin may also promote appetite and increase food intake. A study conducted on healthy volunteers showed that food intake was significantly increased in the intravenous hunger hormone group. The appetite-enhancing effect of ghrelin was associated with increased expression of AgRP mRNA and decreased expression of POMC mRNA.

#### 4.1.3. The Role of Leptin in Central Appetite Regulation

Leptin is mostly derived from white adipose tissue and has a wide range of biological effects. Its main role is to suppress appetite in the hypothalamus, increase energy expenditure and inhibit fat synthesis. Abnormal adipose tissue function in leptin-deficient mice, secondary to overeating and reduced energy expenditure, leads to obesity. Leptin receptors are widely present in the hypothalamus, hippocampus, and other central nervous systems and peripheral organs [57]. Depending on the site of binding of nonreceptor type tyrosine kinase 2 (Janus kinase 2, JAK2) to the leptin receptor protein, its signaling pathway can be divided into the JAK2/signal transducers and activators of transcription 3 (STAT3) pathway, mitogen-activated protein kinase (MAPK)/extracellular signal-regulated kinase (ERK) pathway, and insulin receptor substrate (IRS)/phosphatidylinositol-3-kinase (PI3K) pathway [58].

#### 4.1.4. The Role of Insulin in Central Appetite Regulation

Insulin was first discovered 100 years ago by Banting and Best in extracts from the dog pancreas. Insulin is considered to be a peripheral regulator of blood glucose levels and is used to treat type 1 and type 2 diabetes in more than 450 million people worldwide [59]. Elevated blood glucose levels stimulate the release of insulin, which is circulated in the bloodstream and binds to insulin receptors on the cell membranes of tissues, such as the liver, muscle, and fat. In addition, despite early assertions that the central nervous system is insensitive to insulin, it has recently been discovered that insulin is also synthesized and secreted by nerve cells in the brain. Cortical glial cells may be one of the sources of insulin in the brain [60], and insulin levels in the brain are much higher than plasma insulin levels. Peripheral insulin can also be transferred to the brain via the blood–brain barrier. In the hypothalamus, insulin is involved in the regulation of glucose homeostasis, central glucose transport, appetite, and metabolism [61]. Classic central insulin signaling pathways include the hypothalamic IR/IRS/PI3K/AKT/STAT3 pathway and the IR/IRS/PI3K/ATP pathway.

### 4.2. Central Regulatory Mechanism of Appetite by Sleep Deprivation in Animal Models

Sleep deprivation contributes to the central regulation of appetite by modulating the expression and function of appetite-related hormones [62]. Therefore, we discuss different hormones separately to explore their roles and underlying mechanisms in sleep deprivation-mediated appetite regulation (Figure 2).

#### 4.2.1. Orexin and Ghrelin

Glutamate transporter-1 (GLT1) is an astrocytic transporter protein that is responsible for glutamatergic transmission in the brain. Sleep deprivation significantly alters the localization of glutamate transporter-1 and the excitability of orexin neurons [63]. Specifically, after sleep deprivation, GLT1 was found to be reduced around the cytosol of orexin neurons, leading to different forms of synaptic plasticity [63]. Another rat model also demonstrated that sleep deprivation increased the expression of orexin and that increased expression of orexin was able to inhibit signaling in the ERK1/2 pathway [64]. Although the mechanisms by which sleep deprivation regulates appetite and hunger hormone expression are currently unknown, available studies also suggest that it may affect appetite by influencing appetite neuronal excitability as well as hunger hormone levels [65].

#### 4.2.2. Leptin

In one research study on sleep deprivation in rats, the mRNA levels of leptin receptor (LepRb) in the prefrontal cortex were found to be decreased as compared to the control group by PCR, while the mRNA levels of leptin receptors in the hypothalamus were significantly increased [66]. Furthermore, a study by our team elucidated the biological regulation of leptin receptors in a rat model through the JAK2/STAT3 signaling pathway [26]. Additionally, another study investigating the treatment of appetite suppression with amphetamine (AMPH) found that LepRb/JAK2/STAT3 signaling in the hypothalamus was involved in the regulation of appetite by AMPH [67]. These results showed that sleep deprivation may act as a regulator of appetite-controlling by inducing leptin receptor expression and compromising the JAK2/STAT3 signaling pathway. However, current studies lack in vivo and ex vivo experiments to directly confirm the existence of such mechanisms. Notably, a growing number of studies suggest that circadian rhythms can control energy metabolism [68]. The circadian rhythm, also known as the biological clock, is prevalent in the biological world and is a physiological phenomenon with an approximately 24-h cycle. The circadian system organizes metabolism, physiology, and behavior in a daily circadian cycle that includes a central pacemaker in the brain and a series of clocks in peripheral tissues throughout the body, including the liver, muscle, and adipose tissues [69,70,71,72]. Epidemiological studies have shown that sleep deprivation, shift work, and jet lag syndrome can all cause circadian clock disruption [73,74]. Our in vivo experiment found that sleep deprivation reduced the expression of circadian clock genes in the hypothalamus of rats [26]. Can sleep deprivation affect hypothalamic feeding by modulating changes in circadian rhythms? The results of Kettner et al. also showed that circadian rhythms enhanced the LepRb response to serum leptin within the ARC, conversely, chronic circadian rhythm disturbances led to LepRb desensitization to circulating leptin, resulting in leptin resistance [75]. Together, the above studies suggest that sleep deprivation may regulate appetite by inducing disturbances in biological rhythms to decrease the sensitivity of LepRb to circulating leptin. Similarly, future studies need to further demonstrate the existence of the above mechanisms by establishing sleep deprivation models and interfering with the expression of biological clock genes on this basis.

#### 4.2.3. Insulin

Animal studies have similarly demonstrated that sleep deprivation leads to a reduction of insulin sensitivity through a rhesus monkey model [76]. In another model of paradoxical sleep deprivation (PSD) based on Wistar rats, researchers found an upregulation in food intake and a concomitant decrease in insulin levels during the light phase [77]. Eight Drosophila insulin-like peptides (DILPs) have been identified in Drosophila, and are involved in the regulation of carbohydrate concentrations in the hemolymph and the accumulation of storage metabolites. It was found that insulin-like peptides are able to regulate appetite in Drosophila [78]. Remarkably, this regulatory mechanism also exists in mammals. A study conducted by Chruvattil et al. [79] in Charles Foster rats showed that insulin signaling in the hypothalamus was involved in regulating the feeding behavior of rats, and further studies also found that insulin could regulate appetite in rats by regulating the expression of SIRT1 and activating the AMPK signaling pathway. Although no studies have directly confirmed the involvement of sleep deprivation in the central regulation of appetite through the induction of insulin resistance, we can speculate on the potential of this mechanism and verify it in future scientific experiments.

### 4.3. Central Regulatory Mechanism of Appetite by Sleep Deprivation in Human

Currently, investigations on children and adolescents have shown that sleep deprivation affects serum leptin levels, influencing appetite regulation and leading to weight gain [80]. In a randomized crossover experiment of 19 healthy men under normal sleep and sleep deprivation conditions, researchers took blood samples from volunteers during standardized caloric feeding and found that ghrelin levels increased after sleep deprivation, which may be a source of it [81]. However, there are also studies showing that sleep deprivation is not associated with changes in orexin and ghrelin [82]. These inconsistent results may result from differences in the duration of sleep deprivation, differences in eating conditions during hormone measurements (e.g., standardized versus arbitrary ingestion), or differences in the timing and frequency of blood sampling. The effects of sleep restriction on appetite-regulating hormones may not be detected when measurements are taken during uncontrolled caloric intake. 

In a randomized controlled trial, 10 healthy individuals underwent 4 nights of normal sleep (8 h of bed rest) and 4 nights of sleep deprivation (4 h of bed rest) in a sleep laboratory. Insulin measurements were performed early each morning, and the results showed that the area under the insulin curve was higher in the sleep deprivation experiment than in the control group [83]. Additionally, a narrative review based on clinical evidence reported that sleep deprivation or poor sleep quality was associated with reduced insulin sensitivity [84]. The incretin hormone, glucagon-like peptide 1 (GLP-1) induces satiety and increases postprandial insulin secretion. A study on one-night sleep deprivation in healthy men showed no significant effects on circulating concentrations of total serum GLP-1 but induced temporal changes in the peak GLP-1 response to breakfast intake [85]. The changes in postprandial GLP-1 signaling contribute to the impairment of sleep loss on glucose homeostasis and food intake control, which should be focused on in future studies.

## 5. Discussion and Perspectives

Chronic sleep deprivation has significant adverse effects on health and overall quality of life, and several pieces of data support the hypothesis that reduced sleep duration is associated with an increased risk of being overweight and developing obesity in children. Notably, recent studies suggest that reduced sleep duration can affect children’s ability to self-regulate their appetite, thereby contributing to childhood obesity. These findings have piqued scientists’ interest in exploring the mechanisms by which sleep deprivation regulates appetite. As research has progressed, the establishment of animal models of sleep deprivation has greatly supported researchers to explore the relationship between sleep and food intake, body weight, and metabolism. In recent years, the establishment of an increasing number of models has demonstrated that sleep deprivation contributes to the central regulation of appetite by modulating the expression and function of appetite-related hormones. Existing studies support the possibility that pro-appetitive peptides, including orexin, ghrelin, leptin, and insulin, mediate the central regulation of appetite by sleep deprivation. However, the current study lacks direct evidence that pro-appetitive peptides mediate the central regulation of appetite by sleep deprivation, which provides a new direction for future scientific research. Moreover, the investigation of how sleep deprivation regulates the expression of appetite-promoting peptides and how appetite-promoting peptides affect appetite is mostly phenomenological, and the specific mechanisms still need to be further explored. In addition, recent studies also found that the disruption of circadian clock genes has certain effects on appetite regulation. That means sleep deprivation may interfere with food intake through transcriptional regulation, and the discovery of circadian clock genes and their mechanisms that mediate appetite regulation by sleep deprivation will provide potential targets for clinical interventions in obesity. In conclusion, despite the limitations of the existing evidence on sleep duration and eating behavior, the overall evidence on sleep and obesity suggests that improving sleep may serve as a beneficial intervention for health. In addition, the existing research supports the theoretical basis that it is necessary to further discover the mechanisms by which sleep deprivation regulates appetite, and discovering reliable evidence that supports the existing hypotheses could provide additional guidance for the prevention and protection of the health of the population.

## Figures and Tables

**Figure 1 nutrients-14-05196-f001:**
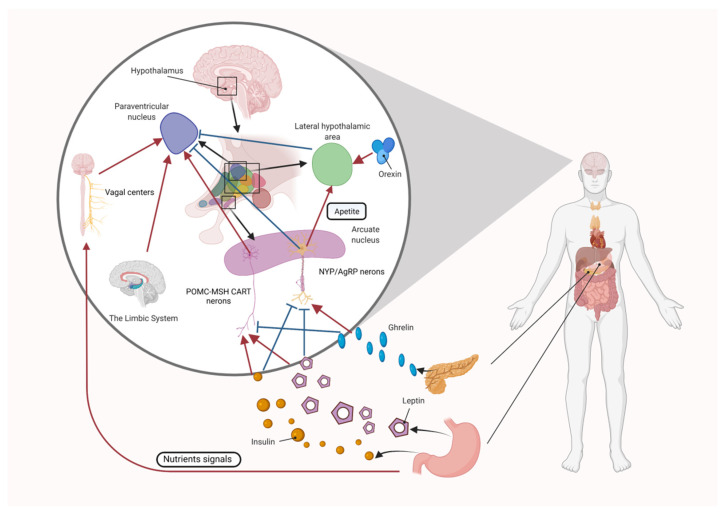
Central regulatory mechanisms of appetite. POMC-MSH neurons and NYP/AgRP neurons in the ARC of the hypothalamus integrate input from the vagus nerve and body fluids, including orexin, ghrelin, leptin, and insulin, and play a physiological role in the regulation of appetite.

**Figure 2 nutrients-14-05196-f002:**
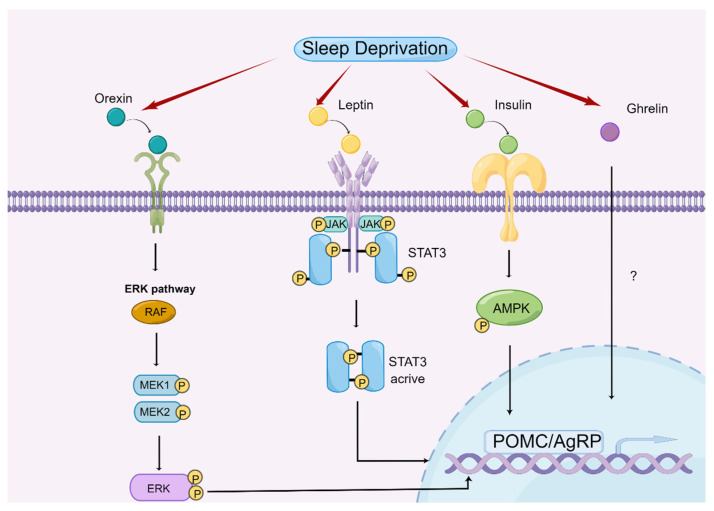
Central regulatory mechanism of appetite by sleep deprivation. Mechanisms of sleep deprivation on the regulation of appetite by affecting orexin, ghrelin, leptin, and insulin (drawn by Figdraw).

**Table 1 nutrients-14-05196-t001:** Animal models of sleep deprivation.

Research Animal	Sleep Deprivation Method	Sleep Deprivation Time	Experiment Time	Food Intake	Weight	Ref.
C57 mouse/1.5 months	Flowerpot	10 h (10:00–20:00)	5 days	Increased	↓	[23]
C57 mouse/7–9 months		10 h (10:00–20:00)			↑	
Wistar rat	Motorized activity wheels	18 h (3 h SD * with 1 h recovery)	5 days	/	↓	[24]
		18 h (3 h SD * with 1 h recovery)	23 days			
Wistar rat	Platform	16 h (7:00–23:00)	8 weeks	Increased	↑	[25]
		16 h (7:00–19:00, 3:00–7:00)				
		16 h (7:00–19:00, 23:00–3:00)				
SD rat	Gentle handling	4 h (8:00–12:00)	4 weeks	Increased	↑	[26]
SD rat	Platform	24 h	3 days	Increased	↓	[27]
C57BL/6J mouse	Platform	24 h	3 days	Increased	↓	[28]
ICR mouse	Rotating drum	24 h	14 days	/	↓	[29]
Wistar rat	Platform	6 h (10:00–16:00)	21 days	/	↓	[30]
		24 h	1 day			
		24 h	4 days			
Wistar rat	Slowly rotating cages	20 h (14:00–10:00)	8 days	Increased ^#^	↓	[31]
SD rat	VLPO- lesioned	/	60 days	Increased	↓	[32]
C57 mouse	Gentle handling	6 h (first six hours of the light phase)	5 days	Increased	↓	[33]
SD rat	platform	1 h (9:00–10:00)	10 days	Increased	↓	[34]

Notes: SD * indicated sleep deprivation. ^#^ indicated the animals were given high-fat diets.

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
