# Peer review of "Sleep Deprivation and Central Appetite Regulation"

_nutrients, 2022, doi:10.3390/nu14245196_

Round 1

Reviewer 1 Report

It is an interesting review. 

the review is interesting and the topic is hot as the population worldwide is becoming both sleep deprived and obese. However, the authors should extract the paragraph about sleep deprivation in animals. The forced movement method is quite different from sleep deprivation in humans and has nothing to add in scientific literature. These experiments are absolutely biased because animals are mainly stressed and as a consequence sleep deprived.

Author Response

Dear Reviewers:

Thank you very much for your time and effort in reviewing our manuscript entitled “Sleep deprivation and central appetite regulation” (Manuscript ID: nutrients-2030170). We appreciate the excellent suggestions and comments by the reviewers. 

Response to Reviewer #1 (Reviewer Suggestions to the Author):

[The review is interesting and the topic is hot as the population worldwide is becoming both sleep deprived and obese. However, the authors should extract the paragraph about sleep deprivation in animals. The forced movement method is quite different from sleep deprivation in humans and has nothing to add in scientific literature. These experiments are absolutely biased because animals are mainly stressed and as a consequence sleep deprived. ]

Response: As the reviewer’s suggestion, sleep deprivation in animals was extracted in paragraph 3.2, and sleep deprivation in humans was extracted separately in paragraph 3.3 in this version of manuscript.

Special thanks to you for your good comments. Revised portions are tracked changes on the paper. Please see the attachment.

Yours Sincerely,

Qi Sun (Corresponding author)

Reviewer 2 Report

The manuscript by Liu et al is a review article dealing with the connections between sleep deprivation and appetite control. The authors make a clear description of the pathophysiological mechanisms linking sleep and regulation of food assumption, particularly focusing on gut-brain and fat-brain axes. They also consider several other aspects involved in appetite regulation.

They provide a comprehensive discussion of cellular, animal and human models, nicely depicting the state of art.

Overall, the manuscript is well organized but only deserves a careful check of the English phrasing. (For example at page 2, "....physiological ...diseases" does not seem correct; a few lines down, "...sleep deprivation" is un-necessarily repeated.)  

Author Response

Dear Reviewers:

Thank you very much for your time and effort in reviewing our manuscript entitled “Sleep deprivation and central appetite regulation” (Manuscript ID: nutrients-2030170). We appreciate the excellent suggestions and comments by the reviewers.

Response to Reviewer #2 (Reviewer Suggestions to the Author):

[Point 1: Overall, the manuscript is well organized but only deserves a careful check of the English phrasing. (For example at page 2, "....physiological ...diseases" does not seem correct.) ]

Response 1: As the reviewer’s suggestion, we have checked the English phrasing for this manuscript and revised this sentence in this version.

[Point 2: a few lines down, "...sleep deprivation" is un-necessarily repeated.) ]

Response 2: As the reviewer’s suggestion, the un-necessarily repeated words “sleep deprivation” have been deleted or replaced by other words in this version of manuscript.

Special thanks to you for your good comments. Revised portions are tracked changes on the paper. Please see the attachment.

Yours Sincerely,

Qi Sun (Corresponding author)

Reviewer 3 Report

Dear authors,

Overall, this is a well written review with good insight into the mechanisms that may control changes in appetite following sleep disruption. I do have some comments for consideration.

Introduction – the introduction is well written, with a good rationale for the review.  Something to consider is reviewing the discussion of the last paragraph on the hypothalamus.  It feels that the discussion has already moved onto possible mechanisms.  Perhaps it might be useful to start more broadly with a discussion that there are many different factors that affect appetite.

Table 1 – It can be difficult to differentiate sleep deprivation time and experiment time.  It may be worth segmenting with lines or shading to make this a little clearer.  I think it would also be useful to mention the type of food the animals were given (e.g. standard chow, high fat diet)

Section 1.3 – I think it would be worth ensuring that the title of this section includes ‘humans’ or ‘children’ to clarify that this section is regarding humans.

Section 2.2.3 – Insulin. This is a very interesting section but I feel this should be expanded upon, perhaps broadened to the effect of incretins, as quite often an interaction between hormones is observed in appetite studies.

Other comments.

There are a number of studies that have investigated circadian misalignment on appetite.  It might be worth considering if these types of studies should be concluded.  I appreciate, that these studies may not provide mechanisms, but do provide proof of concept for this phenomenon  

Author Response

Dear Reviewers:

Thank you very much for your time and effort in reviewing our manuscript entitled “Sleep deprivation and central appetite regulation” (Manuscript ID: nutrients-2030170). We appreciate the excellent suggestions and comments by the reviewers.

Response to Reviewer #3 (Reviewer Suggestions to the Author):

[Point 1: Introduction – the introduction is well written, with a good rationale for the review.  Something to consider is reviewing the discussion of the last paragraph on the hypothalamus.  It feels that the discussion has already moved onto possible mechanisms.  Perhaps it might be useful to start more broadly with a discussion that there are many different factors that affect appetite.]

Response to point 1: We revised the last paragraph of introduction part according to the reviewer’s suggestion.

[Point 2: Table 1 – It can be difficult to differentiate sleep deprivation time and experiment time. It may be worth segmenting with lines or shading to make this a little clearer.  I think it would also be useful to mention the type of food the animals were given (e.g. standard chow, high fat diet) ]

Response to point 2: As the reviewer’s suggestion, sleep deprivation time has been segmented with shading. We have marked with “#” for animals that were given a high-fat diet below Table 1.

[Point 3: Section 1.3 – I think it would be worth ensuring that the title of this section includes ‘humans’ or ‘children’ to clarify that this section is regarding humans. ]

Response to point 3: As the reviewer’s suggestion, the section 1.3 have been moved to section 2 and the title of this section was replaced by “sleep curtailment and appetite regulation in human”.

[Point 4: Section 2.2.3 – Insulin. This is a very interesting section but I feel this should be expanded upon, perhaps broadened to the effect of incretins, as quite often an interaction between hormones is observed in appetite studies. ]

Response to point 4: As the reviewer’s suggestion, we have added this part in section 3.3 about incretins with highlight yellow color in this version of the manuscript.

[Point 5: Other comments: There are a number of studies that have investigated circadian misalignment on appetite. It might be worth considering if these types of studies should be concluded.  I appreciate, that these studies may not provide mechanisms, but do provide proof of concept for this phenomenon.]

Response to point 5: We totally agree the reviewer’s suggestion about circadian misalignment has been involved in appetite regulation. We discussed this field in section 3.3.2 leptin, the hormone which is the most relevant to sleep deprivation and may be regulated by circadian rhythm.

Special thanks to you for your good comments. Revised portions are tracked changes on the paper. Please see the attachment.

Yours Sincerely,

Qi Sun (Corresponding author)
